# AUTO-REGRESSIVE SURFACE CUTTING

## ABSTRACT

Surface cutting is a fundamental task in computer graphics, with applications in UV parameterization, texture mapping, and mesh decomposition. However, existing methods often produce technically valid but overly fragmented atlases that lack semantic coherence. We introduce SeamGPT, an auto-regressive model that generates cutting seams by mimicking professional workflows. Our key technical innovation lies in formulating surface cutting as a next token prediction task: sample point clouds on mesh vertices and edges, encode them as shape conditions, and employ a GPT-style transformer to sequentially predict seam segments with quantized 3D coordinates. Our approach achieves good performance on UV unwrapping benchmarks containing both manifold and non-manifold meshes, including artist-created, and 3D-scanned models. In addition, it enhances existing 3D segmentation tools by providing clean boundaries for part decomposition.

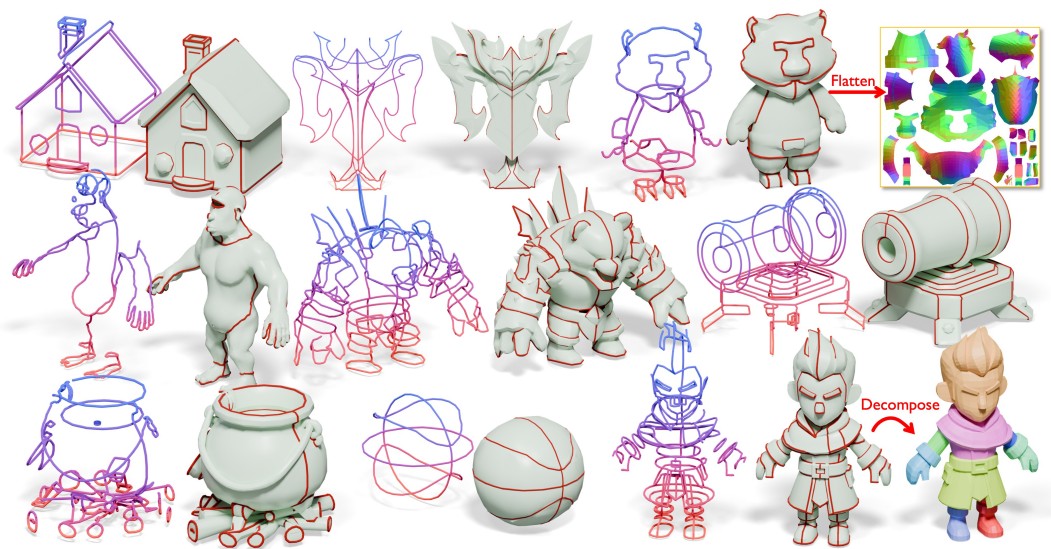

Figure 1: SeamGPT generates surfaces cutting seams, facilitating UV flattening and part decomposition.

## 1 INTRODUCTION

Surface cutting – the process of decomposing 3D mesh geometry through topological incisions – serves as a fundamental operation for numerous applications, including UV parameterization, texture mapping, and digital fabrication. This technique also enables critical downstream tasks such as remeshing, part-based editing, and shape decomposition.

The primary objective of surface cutting is to achieve low-distortion surface flattening. Current industry-standard tools (e.g., XAtlas or Blender's Smart UV Project) optimize for minimal distortion when flattening meshes into 2D atlases. While these methods produce technically valid cuts, they often generate over-fragmented atlases lacking semantic coherence, significantly limiting their utility for tasks requiring meaningful part decomposition (e.g., texture map editing). Nuvo (Srinivasan

et al., 2024) and FAM (Zhang et al., 2024) attempt to improve the continuity of atlases through neural field-based parameterization. However, these methods still struggle to produce globally coherent cuts that align with semantic or functional boundaries. Recent learning-based surface segmentation methods, such as PartField (Zhu et al., 2023), can provide semantic guidance for part decomposition but often fail to produce clean boundaries for cutting. Alternatively, existing wire-frame detection methods (Gori et al., 2017; Zhu et al., 2023; Ma et al., 2025) are limited to simple scenarios (e.g., CAD models).

*Where to put the cuts?* The optimal cuts should 1) minimize geometric distortion for UV parameterization, 2) preserve semantically meaningful boundaries, and 3) maintain functional continuity for downstream applications. In professional practice, artists often manually define mesh cuts in interactive workflows: they initialize cutting at some topological singularities, and gradually connect more points to form cutting seams in a sequential way, and prioritize functionally significant boundaries. The recent release of large-scale 3D artistic mesh datasets (e.g., Objaverse (Deitke et al., 2022) and Objaverse-XL (Deitke et al., 2023)) provides extensive artist-created cutting seam annotations. This motivates us to learn the distribution of artist-crafted seams to automate semantically meaningful surface cutting.

Auto-regressive modeling has achieved remarkable success in content generation across different modalities, such as Large Language Models (LLMs) (Achiam et al., 2023), image generation (Han et al., 2024), and polygon mesh generation (Hao et al., 2024a). We find that auto-regressive modeling is well-suited to the surface cutting task. It mimics the casual decision-making process of manual cutting by capturing the sequential dependencies between successive cuts.

To this end, we introduce SeamGPT, an auto-regressive model that generates artist-style cutting seams, as sequences of 3D line segments. Specifically, given an input surface mesh, we sample point clouds on vertices and edges and compress them into a latent shape condition using a point cloud encoder. A GPT-style transformer decoder then auto-regressively generates cutting seams in the form of line segments. Each seam segment is represented by a start and an end 3D point with quantized coordinates, produced in axis-sorted order. We train SeamGPT on a curated dataset of 560K artist-cut meshes, filtered to retain only those with artist-annotated seams exhibiting clear semantic intent. Experiments across multiple mesh categories demonstrate that our method significantly improves surface cutting quality for UV-unwrapping. In addition, in certain scenarios, the generated seams can serve as effective boundary indicators for 3D part segmentation, providing a potential solution for advancing 3D segmentation.

Our key contributions are as follows:

- We present a new auto-regressive formulation for cutting seams as a sequence of line segments, tailoring a GPT-inspired transformer, to produce semantically meaningful and consistent artist-style cutting.
- For UV-unwrapping, SeamGPT achieves state-of-the-art performance on benchmarks containing both manifold and non-manifold meshes, including artist-created meshes and AI-generated meshes (e.g., from latent 3D diffusion models and polygon-generation methods).
- For part segmentation, SeamGPT complements existing tools (e.g., Parfield (Liu et al., 2025)) by providing clean boundaries, addressing a key limitation in current boundary refinement pipelines.

## 2 RELATED WORK

**Mesh Cutting for Flattening.** Mesh cutting represents a well-established research problem in computer graphics. The primary use of mesh cutting is to achieve low-distortion mesh flattening for texture mapping. Cutting strategies are broadly categorized as bottom-up or top-down. Bottom-up approaches first partition the surface into small, compact regions and iteratively merge connected regions (Sorkine et al., 2002; Zhou et al., 2004; Yamauchi et al., 2005). Sorkine et al. (Sorkine et al., 2002) pioneered this paradigm by expanding seed triangles until reaching distortion bounds, later refined in (Zhou et al., 2004; Yamauchi et al., 2005). These methods form the foundation for industrial tools like xAtlas and Blender's Smart UV Unwrapper. While technically valid, they often produce over-fragmented atlases lacking semantic coherence, which limits their utility for texture editing workflows. Top-down methods cut the surface with seams or winding boundaries, either through

user-guided or automatic techniques (Mitani & Suzuki, 2004; Tang et al., 2016; Sheffer & Hart, 2002). Cutting seams can be automatically generated by connecting cone singularities (Kharevych et al., 2006; Springborn et al., 2008; Soliman et al., 2018). After defining seams, the flattening process typically relies on distortion optimization techniques based on mesh connectivity. Least Squares Conformal Mapping (LSCM) (Lévy et al., 2023) is widely used for generating conformal maps for pre-cut meshes. Variational Surface Cutting (Sharp & Crane, 2018) presents a global variational approach for low-distortion cuts, using conformal shape derivative flow for parameterization-free flattening with user constraints. Geometry Images (Gu et al., 2002) and Multi-Chart Geometry Images (Sander et al., 2003) encode 3D meshes as regular grids for efficient storage and re-meshing. Rectangular Multi-Chart Geometry Images (Carr et al., 2006) enabled GPU-friendly regular grids. OptCuts (Li et al., 2018) jointly optimize cut length and distortion through alternating mapping and cut locus optimization; the process involves hybridized discrete/continuous optimization. SeamCut (Lucquin et al., 2017) automates segmentation via field-based analysis while remaining topology-agnostic. Recent neural field-based approaches leverage continuous MLP functions to model cutting and UV mapping. AtlasNet (Groueix et al., 2018) pioneered this direction by using neural fields to represent a continuous mapping from 2D atlases to 3D surfaces. Nuvo (Srinivasan et al., 2024) optimizes multiple neural fields for UV unwrapping with explicit parameterization constraints. FAM (Zhang et al., 2024) extend this direction with interpretable sub-networks surface cutting, UV deforming, unwrapping, and wrapping, the sub networks are assembled into a bi-directional cycle mapping framework. While these methods generate continuous mappings, they typically require per-scene optimization and often disregard object semantics.

**3D Part Segmentation.** Traditional data-driven part segmentation approaches require predefined part templates, follow a supervised learning pipeline, and suffer from the limited scale and diversity of 3D part-annotated datasets. Due to the success of 2D foundation models like SAM (Kirillov et al., 2023), recent open-world 3D segmentation has made significant progress. Various methods have been developed to lift and merge multi-view 2D SAM predictions for both scene-level and object-level 3D segmentation. For instance, Part123 (Liu et al., 2024) integrates SAM into 3D reconstruction, enabling part-aware single-image object generation. Similarly, SAMPart3D (Yang et al., 2024) employs per-shape optimization to distill 2D segmentations into 3D. However, these approaches rely on multi-step inference pipelines with lengthy optimizations. In contrast, PartField (Liu et al., 2025) proposes a single-stage feedforward method for learning part-based 3D features. HOLOPart (Yang et al., 2025) further refines segmented parts using a 3D diffusion model that captures both part geometry and global shape information—though the completion quality heavily depends on the initial segmentation. A key limitation of existing 3D part segmentation methods is their inability to produce clean part boundaries. In this paper, we demonstrate that our auto-regressive surface-cutting approach has the potential to alleviates this issue by providing precise cutting seams as clean part boundaries.

**Line and wireframe reconstruction.** Reconstructing lines or wireframes from input point clouds or meshes is a closely related task. Traditional line feature detection in point clouds primarily relies on local geometric features, such as the eigen-structure of the covariance matrix, normals, curvatures, or statistical metrics (Bazazian et al., 2015; Daniels et al., 2007; Lin et al., 2015; Hackel et al., 2016). With the advent of deep learning, several works (Wang et al., 2020; Bazazian & Parés, 2021; Himeur et al., 2021) leverage neural networks to detect edge point and formulate it as a per-point classification task. EC-Net (Yu et al., 2018) reformulates this as a regression problem, learning residual point coordinates and point-to-edge distances to identify edge points. PointNet (Qi et al., 2017) and 3D graph convolutions (Wang et al., 2019) are widely adopted as baselines for point feature detection. The next step extracting parametric curves from detected points or lines. As done in (Gumhold et al., 2001; Wang et al., 2020), a common strategy is grouping and connecting the detected points, then determining the target parametric curve type for fitting. NerVE (Zhu et al., 2023) leverages piece-wise linear (PWL) curve representations to enable efficient parametric curve extraction.

## 3 DATA PREPARATION

**Data collection and filtering.** We collected our training data from several open-source 3D datasets, including Objaverse (Deitke et al., 2022), Objaverse-XL (Deitke et al., 2023), and 3D-FUTURE (Fu et al., 2021), focusing primarily on meshes with existing UV coordinates that contain essential

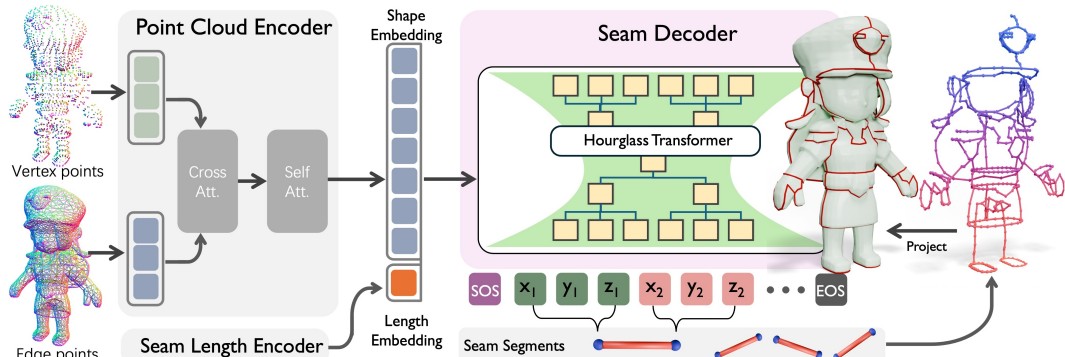

Figure 2: SeamGPT architecture: Point cloud encoder extracts shape context; Causal transformer decoder generates axis-ordered seam coordinates. Color indicates the prediction order is of the seam segments (red to blue).

artist-defined seam information. To ensure high-quality training data, we implemented a rigorous filtering process that excluded meshes with poor topology such as raw 3D scans and low-quality AI-generated models. We removed problematic UV unwrappings such as those with overlapping UVs or arbitrarily placed seams lacking semantic meaning. Through this comprehensive filtering process, we assembled a training dataset of 1.9 million objects for initial training, then further refined our approach by fine-tuning on a high-quality subset of approximately 560K meshes featuring exemplary artist-created seams.

**Mesh Seam Extraction.** To extract ground-truth cutting seams from our filtered meshes, we leveraged the existing UV parameterization information. The boundaries between UV islands naturally define the cutting seams on the original mesh. We identified all UV island boundary edges and projected them back onto the 3D mesh to obtain the corresponding cutting seams. These edges were then converted into our sequential representation format as ordered sequences of line segments.

## 4 METHOD

We introduce SeamGPT, a novel framework that generates artist-style cutting seams through an auto-regressive approach. Our method formulates surface cutting as a sequence prediction problem, where cutting seams are represented as an ordered series of 3D line segments. Given an input mesh $M$, our goal is to generate seam edges $S = \{s^i\}_{i \in [N_s]}$. The overview of SeamGPT is shown in Fig. 2. We first introduce our seam representation strategy in Sec. 4.1, which encodes cutting seams as sequential tokens. In Sec. 4.2, we detail our auto-regressive generation process, which mimics the sequential decision-making of professional artists.

### 4.1 MESH SEAM REPRESENTATION

A seam sequence $S$ of $N_s$ segments $[s^i]_{i \in [N_s]}$ is defined as: $S = [s^1, s^2, \ldots s^{N_s}]$, where each segment $s^i$ is a 3D line segment represented by two vertices: $s^i = (p_h^i, p_t^i)$, i.e. head and tail. Each vertex $p$ is defined by its 3D coordinates: $p = (x, y, z)$. Thus, a seam sequence can be decomposed at multiple levels:

$$
\begin{aligned}
S &= [s^1, s^2, \ldots s^{N_s}] & \text{Segment level} \\
&= [p_h^1, p_t^1, p_h^2, p_t^2, \ldots, p_h^{N_h}, p_t^{N_h}] & \text{Point level} \\
&= [x_h^1, y_h^1, z_h^1, x_t^1, y_t^1, z_t^1, \ldots, x_t^{N_s}, y_t^{N_s}, z_t^{N_s}] & \text{Coord. level}
\end{aligned}
\tag{1}
$$

**Seam ordering.** For an auto-regressive model to function properly, a consistent order of sequences is required. Following existing practice for mesh generation (Siddiqui et al., 2023; Weng et al., 2024; Hao et al., 2024a), we first sort vertices $yzx$ order, where $y$ represents the vertical axis, and then sort two vertices within an edge lexicographically, placing the lowest $yzx$-ordered vertex first. Finally, seam edges are sorted in ascending $yzx$-order based on the sorted values of their vertices. The

resulting order can be seen through the color coding of the generated meshes presented in Figure 2, i.e. from red to blue.

**Quantization of coordinates.** Autoregressive models typically sample from a multinomial distribution over a discrete set of possible values. To adhere to this convention, we quantize vertex coordinates into a fixed number of discrete bins. The quantization resolution—determined by the number of bins—directly affects the precision of the predicted seam. Higher quantization levels yield more detailed and accurate representations but also increase the complexity of the generation process. To balance precision and tractability, we employ 1024-level quantization, enabling effective representation of complex seams.

## 4.2 Autoregressive Seam Prediction

In autoregressive seam prediction, a seam sequence $S$ is generated by sequentially predicting each coordinate $c_i$ based on its conditional probability given all previously generated coordinates $P(c_i|c_{<i})$. The probability of the entire seam is then given by the joint probability of all its coordinates:

$$P(S) = \prod_{i=1}^{6N_s} P(c_i|c_{<i}). \tag{2}$$

**Global Shape Conditioning.** Point clouds are a flexible and universal 3D representation that can be efficiently derived from other 3D formats, including meshes. We use a point cloud encoder to extract representative features for characterizing the input 3D shapes. In the context of surface cutting, seams are encouraged to align with the vertices and edges of the original mesh, such that cutting the mesh along seams does not create excessive extra faces. To guide the decoder in producing vertex and edge-aligned seam placement, instead of sampling point clouds uniformly, we sample surface points only on vertices and along edges. Specifically, we sample a total of 61,440 points, evenly split between: 30,720 points on vertices and 30,720 points on edges. If the input mesh has fewer than 30,720 vertices, we use repeated over-sampling. Points along an edge are sampled uniformly by interpolating between its start and end points with K samples, where K is determined based on the edge's length, where K is computed via $\lfloor 30,720 * \alpha \rfloor$, where $\lfloor \rfloor$ is the rounding function and $\alpha$ is the ratio to the total edge length . Finally, the input points are fed into a jointly trained point cloud encoder from (Team, 2025), which processes the point cloud through a series of cross- and self-attention layers and compresses the point cloud to a latent shape embedding of length 3072 and dimension 1024. Another option to create shape embeddings is to use mesh encoders, such as (Zhou et al., 2020). However, the computational cost of mesh encoder does not scale well when the input has a large number of vertices. We show in the ablation study that point cloud conditioning produces much better results than mesh conditioning.

**Seam Count control.** Given an input shape, multiple possible suitable cutting solutions exist. Depending on the application requirements, one can make many cuts to decompose a mesh, or just a few. To regulate the cutting granularity, we concatenate a length embedding to the shape embedding. We find that modulating the length embedding directly controls cutting granularity.

**HourGlass Decoder Architecture.** Following (Hao et al., 2024a), we build an hourglass-like autoregressive decoder architecture to sequence at multiple levels of abstraction. The architecture employs multiple Transformer stacks at each level, with transitions between levels managed by causality-preserving shortening and upsampling layers that bridge these hierarchical stages. There are three hierarchical levels: coordinates, vertices, and edges. The input coordinate sequence is shortened by a factor of 3 at the vertex level. It is then further shortened by a factor of 2 at the edge level. Both shortening and upsampling layers are implemented to preserve causality. The expanded sequence is combined with higher-resolution sequences from earlier levels via residual connections, similar to U-Nets.

**Training Strategy.** We employ two loss functions to for model training: a cross-entropy loss for token prediction and a KL-divergence loss to regularize the shape embedding space, ensuring it remains compact and continuous. Training begins with a 2,000-step warm-up phase and is parallelized across 64 Nvidia H20 GPUs (96GB Mem.) with a total batch size of 128. The model converges after one week of training. During training, we first scale all samples to fit within a cubic bounding box in the range of $-1$ to $1$. We then apply data augmentation techniques, including random scaling within $[0.95, 1.05]$, random vertex jitter, and random rotation.

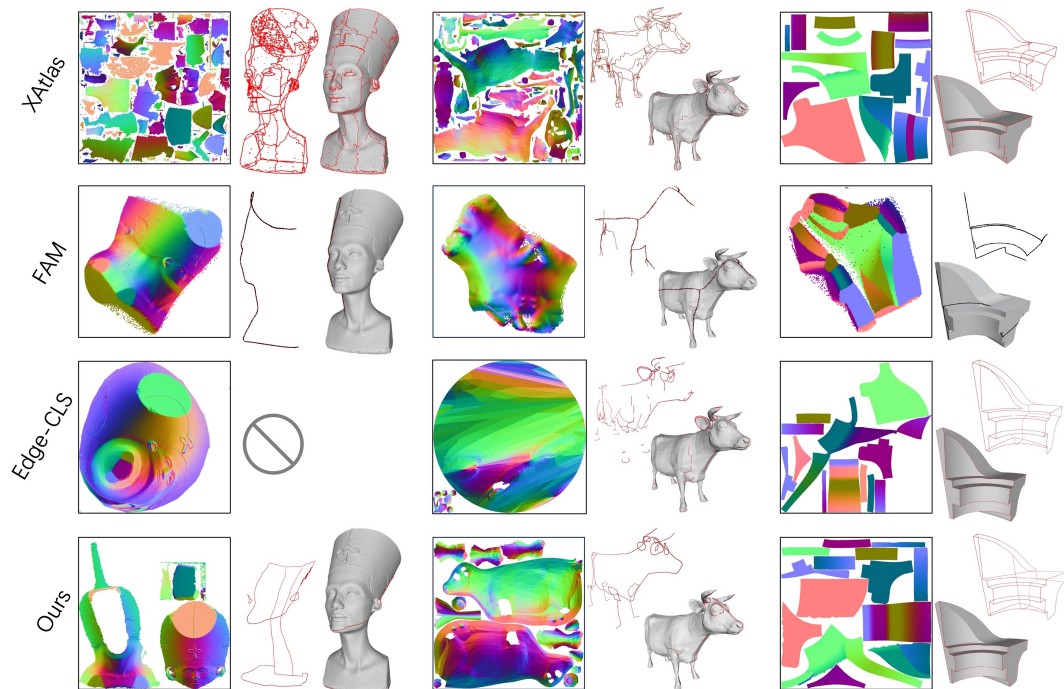

Figure 3: Qualitative UV flatten results on FAM benchmark ( Nefertiti, Cow, and Fandisk). XAtlas and FAM are single-stage algorithms: they directly output the UV-Flatten result. Edge-CLS and Ours are two-stage algorithms: (1) predicting seams on the surface first, and (2) flattening with Blender's minimum stretch.

## 5 MESH UV-UNWRAPPING EXPERIMENT

**Benchmarks and Evaluation Metric.** We conduct experiments on a diverse collection of 3D surface models from Flatten Anything (FAM) (Zhang et al., 2024), which primarily includes low-poly meshes, CAD models, and 3D scanned meshes. We also evaluate on Toys4K (Stojanov et al., 2021), a dataset of non-manifold artist-created meshes. Our evaluation leverages the **Mesh distortion** metrics, which is computed as the average conformal energy over all triangular faces of the mesh.

Table 1: Compare with Xatlas on 105 toy4K objects. SeamGPT yield lower distortion with fewer fragments.

| Method | XAtlas | SeamGPT |
|---|---|---|
| Runtime(s) | **8.8** | 19.1 |
| #. Fragments($\downarrow$) | 1110 | **153** |
| Distortion($\downarrow$) | 10.05 | **6.08** |

**Baselines and Implementations.** We compare SeamGPT against several state-of-the-art methods for mesh UV-unwrapping. **XAtlas** (Young, 2024) employs a bottom-up approach with bounded distortion charts. **Nuvo** (Srinivasan et al., 2024) leverages neural fields with explicit parameterization constraints. **FAM** (Zhang et al., 2024) implements interpretable sub-networks in a bi-directional cycle mapping framework. We also built another baseline called **Edge-CLS**, which takes a mesh as input and uses graph convolution and Transformer layers to compute per-edge features. These features are then fed into an MLP classifier to predict whether each edge is a seam edge or not (i.e., this is an edge classification baseline). We train Edge-CLS on the same training set and use the same UV-unwrapping process as SeamGPT.

**SeamGPT-based UV-unwrapping.** Once SeamGPT generates cutting seams, we implement a streamlined unwrapping process to create practical UV maps. We first map each predicted seam point to its nearest vertex on the input mesh, then connect these vertices through shortest geodesic paths along the mesh edges. We then cut the mesh by duplicating vertices along these paths, creating independent boundaries for flattening. Finally, we apply Blender's Minimum Stretch algorithm to the segmented mesh, optimizing UV coordinates to evenly distribute stretching while preserving

the semantic structure defined by our seams. This process yields low-distortion UV mappings that respect functional and aesthetic boundaries, improving upon conventional automated methods.

**Comparison results.** Results. Tables 2 and 3, along with Figure 3, present qualitative and quantitative results. SeamGPT achieves the best performance across all metrics. In contrast: XAtlas generates over-fragmented cuts, FAM fails to produce subtle cuts consistently, Edge-CLS performs well only on sharp edge features but struggles with generating seams on smooth, featureless regions. Our method consistently produces semantic and reasonable cuts regardless of surface characteristics.

Table 2: Quantitative results on Flatten-Anything benchmark using the face distortion metric.

| | Bimba | Lucy | Ogre | Armadi. | Bunny | Nefert. | Dragon | Homer | Happy | Fandi. | Spot | Arm | Cow | Avg. |
|---|---|---|---|---|---|---|---|---|---|---|---|---|---|---|
| Xatalas (Young, 2024) | 15.44 | **0.01** | **0.66** | **0.17** | 61.84 | **0.03** | **0.22** | **7.51** | 99.84 | 8.41 | 12.77 | 29.98 | **1.94** | 18.37 |
| Nuvo (Srinivasan et al., 2024) | 19.12 | 57.89 | 26.22 | 114.21 | 16.84 | 20.92 | 61.03 | 21.92 | 267.43 | 19.76 | 12.93 | 37.34 | 12.70 | 52.95 |
| FAM (Zhang et al., 2024) | 12.10 | 35.14 | 11.55 | 59.87 | 7.33 | 11.21 | 904.89 | 14.19 | 23.00 | 12.21 | 9.37 | 20.98 | 8.49 | 86.95 |
| Edge-CLS | **8.14** | 22.85 | 23.54 | 11.18 | **3.91** | 4.67 | 15.39 | 20.16 | **27.37** | 12.47 | **5.77** | 87.20 | 9.20 | 19.37 |
| Ours | 10.68 | **0.01** | 2.01 | 2.47 | 50.47 | 0.12 | 0.56 | 10.28 | 61.68 | **8.15** | 5.95 | **14.88** | 2.24 | **13.04** |

Table 3: Quantitative results on Toys4K Benchmark using the face distortion metric.

| | Bowl | Ball | Sheep | Driver | Chicken | Apple | Giraffe | Bottle | Avg. |
|---|---|---|---|---|---|---|---|---|---|
| Xatalas (Young, 2024) | 0.91 | **0.26** | **1.19** | 4.61 | 2.36 | **3.11** | 2.85 | **0.57** | 1.98 |
| Nuvo (Srinivasan et al., 2024) | 3.99 | 1.33 | 10.43 | 33.07 | 9.79 | 15.39 | 21.04 | 6.02 | 12.63 |
| FAM (Zhang et al., 2024) | 3.80 | 0.81 | 6.45 | 15.33 | 18.98 | 6.64 | 11.77 | 4.36 | 8.52 |
| Ours | **0.49** | 0.31 | 1.39 | **4.25** | **1.86** | 4.02 | **2.59** | 0.67 | **1.95** |

**User study.** To further assess our method's practical utility, we conducted a user study with 20 professional 3D artists evaluating **Boundary** quality and **Editability**. Boundary quality measures how unfragmented a UV map is, while editability reflects how well the mapping supports appearance editing. Participants rated UV unwrappings from all methods on a 5-point scale. As shown in Table 4, SeamGPT significantly outperforms existing methods in both metrics.

Table 4: User Study about Boundary quality and Editability.

| | Fandisk | | Cow | | Nefertiti | | Avg. | |
|---|---|---|---|---|---|---|---|---|
| | Boundary ↑ | Editability ↑ | Boundary ↑ | Editability ↑ | Boundary ↑ | Editability ↑ | Boundary ↑ | Editability ↑ |
| Xatalas (Young, 2024) | **4.42** | **4.42** | 2.68 | 2.37 | 2.79 | 2.47 | 3.30 | 3.09 |
| Nuvo (Srinivasan et al., 2024) | 1.32 | 1.32 | 1.16 | 1.21 | 1.42 | 1.42 | 1.30 | 1.32 |
| FAM (Zhang et al., 2024) | 1.74 | 1.53 | 1.84 | 1.53 | 2.05 | 1.84 | 1.88 | 1.63 |
| Edge-CLS | 3.89 | 3.84 | 2.79 | 2.32 | 2.58 | 2.16 | 3.09 | 2.77 |
| Ours | 4.37 | 4.32 | **4.16** | **4.16** | **3.47** | **3.58** | **4.00** | **4.02** |

# 6 SEAM ENHANCED 3D PART SEGMENTATION

In this section, we explore the potential of our SeamGPT to enhance part segmentation. 3D part segmentation is a long-standing problem in 3D computer vision, aiming to separate a 3D object into meaningful components. Existing part segmentation methods such as SAMPart3D (Yang et al., 2024) and PartField (Liu et al., 2025) offen produce blury part boudaries. We propose utilizing the seam lines generated by SeamGPT to complement existing tools and provide clean boundaries for part segmentation.

**Patch-based Part Segmentation.** Given a 3D shape $X$, we first utilize PartField to obtain the initial part segmentation, resulting in a mapping of each face to a part label:

$$L : F \to \mathcal{L}$$

where $F$ is the set of faces and $\mathcal{L}$ is the set of part labels. For each face $f_i \in F$, the function $L$ maps it to a specific label $l_j \in \mathcal{L}$. Then we employ SeamGPT to predict the 3D seam lines, partitioning the shape into multiple patches $P_k$ by connected components :

$$X \to \{P_1, P_2, \ldots, P_n\}$$

For each patch $P_k$, we calculate the count of faces associated with different part labels:

$$C(l_j) = \sum_{f_i \in P_k} \mathbb{I}(L(f_i) = l_j)$$

We select the part label $l_j^*$ with the highest count as the label for the patch $P_k$:

$$l_j^* = \arg\max_{l_j \in \mathcal{L}} C(l_j)$$

Through this Patch-based Part Segmentation methodology, we can obtain well-defined parts with clean boundaries. As demonstrated in Figure 6, our approach significantly improves the accuracy of part decomposition, enhancing the clarity and interpretability of the segmented 3D shape.

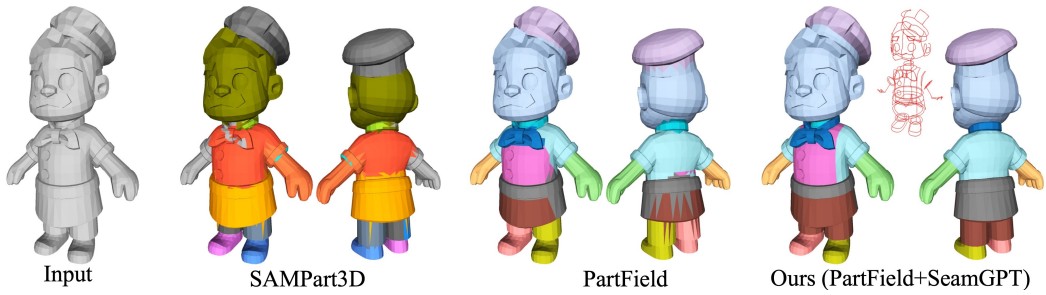

Input       SAMPart3D       PartField       Ours (PartField+SeamGPT)

Figure 4: Qualitative part segmentation results. Our method enhances existing part segmentation methods by providing clean part boundary guidance.

# 7 ABLATION STUDY AND NETWORK DESIGN CHOICES

**Point cloud sampling strategy.** As shown in Figure. 5, when conditioned on point clouds uniformly sampled across the mesh surface, the generated seams remain logically valid from a surface-cutting perspective but may not precisely align with the input mesh's vertices and edges. In contrast, sampling point clouds along edges and vertices produces seams that naturally conform to the mesh topologies. This could prevents creating excessive extra mesh faces. We also found that sampling along edges and vertices significantly improves model convergence, as the transformer gains explicit positional awareness of potential cutting coordinates.

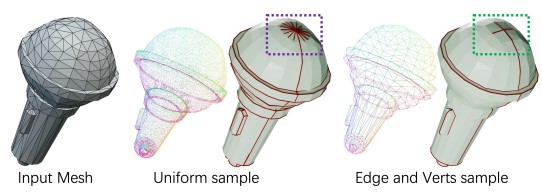

Input Mesh      Uniform sample      Edge and Verts sample

Figure 5: Ablation of point sampling strategy.

**Mesh encoder vs Point cloud encoder.** An alternative approach for generating shape embeddings employs mesh encoders, as demonstrated by Zhou et al. (Zhou et al., 2020). We implemented an encoder combining graph convolutions (operating on both vertices and edges) with a full self-attention transformer (8 SAGEConv layers + 21 Atten. layers) . This encoder produces vertex-wise tokens that are subsequently fed to the decoder via cross-attention mechanisms. As shown in Figure 6, point-cloud encoder yields superior results compared to mesh encoders. Furthermore, the computational cost of our mesh encoder scales poorly with increasing vertex counts. Mesh encoder-based methods often fail to accurately capture the precise positions of original vertices, resulting in significant misalignment between the generated seam edges and the original mesh.

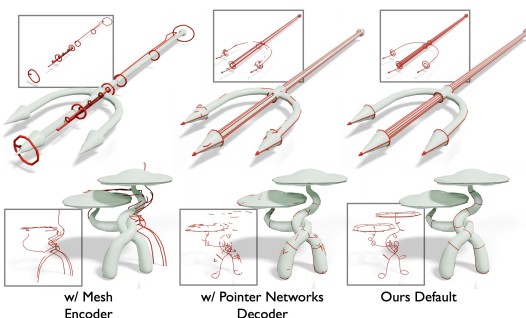

w/ Mesh      w/ Pointer Networks      Ours Default
Encoder      Decoder

Figure 6: Ablation study of encoder and decoder.

**Does Pointer networks works?** In the case that the cutting seam forms a subset of the edges in the mesh, we can also adopt the Pointer Network (Vinyals et al., 2015) architecture, which auto-regressively produce the pointers to the mesh edges. We follow the implementation of Polygen (Nash et al., 2020) to build a pointer network with a mesh encoder that produces edge-wise embedding and a casual transformer to create pointers to the edges that lie on the seams auto-regressively. Pointer network struggles to generate consistent seams, often resulting in discontinuous cuts as demonstrated in Figure 6.

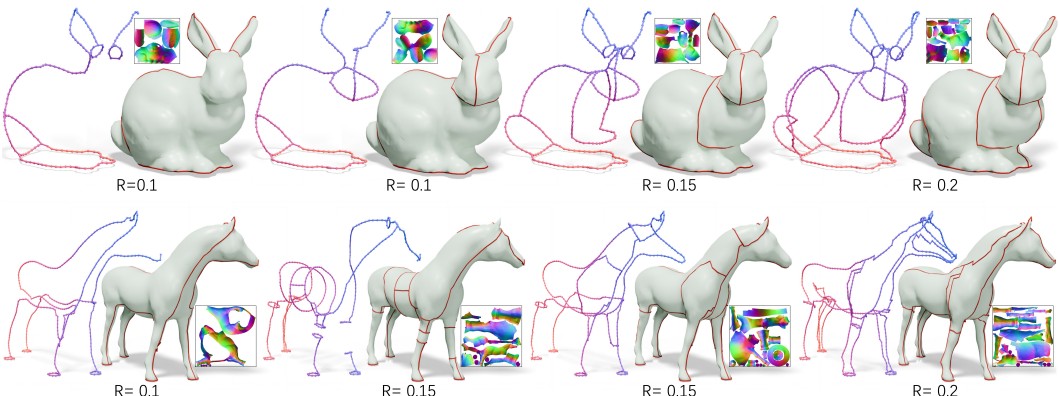

R=0.1        R= 0.1        R= 0.15        R= 0.2

R= 0.1        R= 0.15        R= 0.15        R= 0.2

Figure 7: Seam length control and diversity. We can control the cutting granularity by adjusting seam length. Diverse valid cutting seams can be generated.

**Seam length control and diversity.** We define $R$ as the ratio of seam segment count to the number of mesh vertices. Empirically, valid cutting seams typically have $R$ values within the range $[0.1, 0.35]$. We test $R$ in the range $[0.05, 0.4]$ on the Igea model from the FAM benchmark. As shown in Table 5, distortion generally decreases with increasing $R$, but the improvement slows after $R > 0.3$. The number of fragments steadily increases with $R$, confirming that $R$ values above 0.35 lead to over-fragmentation. For the experiments in the rest of this paper, R is empirically set to 0.2 to balance between distortion and fragment count .

Table 5: Effect of seam length ratio $R$ on distortion and fragment count.

| $R$ | 0.05 | 0.1 | 0.15 | 0.2 | 0.25 | 0.3 | 0.35 | 0.4 |
|---|---|---|---|---|---|---|---|---|
| Distortion | 233.65 | 222.69 | 180.37 | 206.96 | 175.57 | 143.37 | 141.11 | 139.8 |
| Num. of Fragments | 4 | 14 | 9 | 13 | 53 | 41 | 153 | 168 |

As shown in Figure 7, controlling $R$ allows us to adjust the granularity of cuts. Additionally, due to the non-deterministic nature of autoregressive transformers, we can generate diverse valid cutting seams from the same length control.

## 8 CONCLUSION

In this paper, We present SeamGPT, an auto-regressive model that generates artist-style cutting seams for 3D meshes. By formulating surface cutting as a sequence prediction problem, our approach produces semantically meaningful and functionally coherent cuts that outperform existing methods on both UV unwrapping and part segmentation tasks. Our model effectively captures artistic cutting priors, addressing key limitations in current mesh processing workflows and offering a step toward more intelligent and semantically aware tools that better align with artist intentions.

**Limitations.** Our current implementation exhibits degraded performance on meshes exceeding 20K triangular faces, requiring pre-processing via remeshing to reduce face count. This occurs because seam length inherently increases with mesh face count. Potential future directions include: 1) incorporating more powerful transformer decoders to handle complex topologies and longer sequence, and 2) adopting more compact wire representations (e.g., curves) for more efficient seam handling. Additionally, jointly learning part feature extraction and part boundary generation is a promising direction for more robust and generalizable 3D part segmentation.

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

# Appendix

## A  VIDEO

A supplementary video is provided to illustrate the process of generating seams and to present additional generation results produced by SeamGPT.

## B  BROADER IMPACT.

Our paper presents SeamGPT, a learned approach for UV unwrapping and surface cutting. It establishes the foundation for an important research direction in automated 3D geometry processing, which is needed for a variety of applications where high-quality 3D content creation and manipulation is required. These applications range from the fields of game development and digital entertainment to architectural visualization and industrial design. In the former, efficient and high-quality UV unwrapping is of major importance in order to enable realistic texture mapping and material editing. Applications such as virtual world construction and 3D asset generation would greatly benefit from research like ours. This, in turn, could provide society with more accessible tools for 3D content creation, lowering the barrier for artists and designers. On the other hand, as a low-level building block for 3D geometry processing, our work has no direct negative outcome, other than what could arise from the aforementioned applications.

## C  METRICS

We adopt the metrics that evaluates the conformality of the UV parameterization via per-face distortion. Specifically, for each triangle in the 3D mesh, we compute the Jacobian matrix $J \in \mathbb{R}^{2 \times 3}$ that maps the 3D triangle to its flattened 2D UV counterpart via the linear relation:

$$V_{2D} = J \cdot V_{3D}. \tag{3}$$

Here, $V_{3D} \in \mathbb{R}^{3 \times 3}$ and $V_{2D} \in \mathbb{R}^{2 \times 3}$ represent the vertex positions of a triangle in 3D and 2D space, respectively. To quantify the local distortion, we compute the singular values $\sigma_1, \sigma_2$ of the Jacobian matrix $J$. These values capture the local stretching or compression in the $u$ and $v$ directions. Ideally, conformal mappings preserve angles, corresponding to $\sigma_1 = \sigma_2 = 1$. Deviation from 1 reflects distortion.

Following prior works (Park et al., 2021), we compute the conformal energy per triangle as:

$$E_{conf} = |\log \sigma_1| + |\log \sigma_2| \tag{4}$$

and report the mean value across all mesh triangles. This metric effectively penalizes non-uniform scaling and captures both area and angle distortions.

In the FAM paper, this is done by computing the average of the **absolute angle differences (AAD)** between each corresponding angle of the 3D triangles and the 2D parameterized triangles. (See the 3rd line from the bottom on page 7 of the FAM paper (Zhang et al., 2024))

| Model | Bimba | Lucy | Ogre | Armadillo | Bunny | Nefertiti | Dragon | Homer | Happy | Fandisk | Spot | Arm | Cow | Average |
|---|---|---|---|---|---|---|---|---|---|---|---|---|---|---|
| FAM | 0.064 | 0.166 | 0.227 | 0.154 | 0.073 | 0.038 | 0.239 | 0.048 | 0.175 | 0.087 | 0.080 | 0.040 | 0.106 | 0.115 |
| SeamGPT | 0.000 | 0.011 | 0.009 | 0.008 | 0.000 | 0.001 | 0.184 | 0.001 | 0.188 | 0.000 | 0.003 | 0.000 | 0.003 | 0.031 |

Table 6:  Results using the absolute angle differences (AAD) metric from FAM paper.

## D  MORE IMPLEMENTION DETAILS

### D.1  POINT SAMPLING STRATEGY

Our point sampling strategy plays a critical role in guiding the transformer decoder to generate mesh-aligned seams. Instead of uniformly sampling the mesh surface, we explicitly sample points from mesh vertices and edges to ensure that the predicted seams conform to the original mesh topology.

Specifically, for each input mesh, we sample a total of 61,440 points, equally divided between vertices and edges. That is, we sample 30,720 points on vertices and 30,720 points along edges. For edge sampling, we perform uniform interpolation between edge endpoints, where the number of samples per edge is proportional to its length. If the number of vertices is fewer than 30,720, we apply repeated oversampling to reach the desired count.

This targeted sampling improves seam quality in two ways: (1) It prevents generation of seams that do not align with mesh connectivity, reducing the number of extra faces introduced during mesh cutting; and (2) it provides the transformer with more meaningful structural cues, accelerating convergence and improving generation accuracy.

### D.2  Architecture and Training Details

**Transformer Decoder.**  We adopt a hierarchical hourglass-style transformer decoder with causal attention, as in recent mesh autoregressive models (Hao et al., 2024a). The architecture is defined by a three-level abstraction structure with depth configuration `(2, (4, 12, 4), 2)`, where each number represents the number of transformer blocks at that level. Each block has dimension 1536 and uses 16 attention heads with per-head dimension 64. Causal masking is applied at all levels to preserve autoregressive consistency, Fig. 9 shows the architecture.

To facilitate efficient training and decoding of long sequences, we use position encoding for a maximum sequence length of 36,864, with quantized coordinate representation at 10-bit precision ($2^{10} = 1024$ discrete bins).

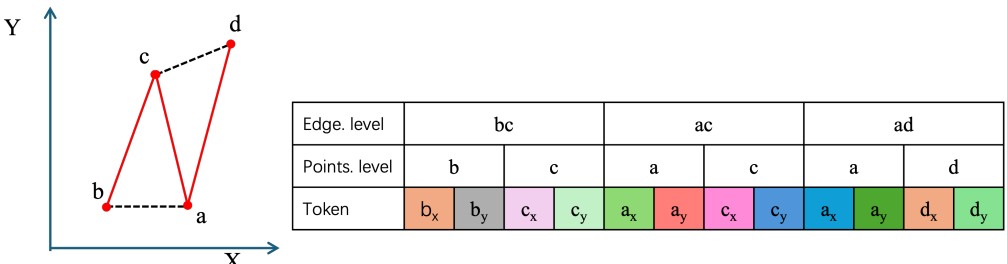

Figure 8:  Illustration of the tokenization process using a 2D mesh example (left). The red edges indicate seam lines. The table on the right shows the token sequence. Points are firsts sorted by this y-x order. Then edge are sorted according to their "earliest" points.

**Sequence Control.**  To regulate the length of generated seams, a length embedding is concatenated to the shape embedding. The maximum truncated length is set to 27,000 tokens, and training is performed in truncated mode to stabilize long-sequence generation.

**Training Setup.**  The model is trained with a batch size of 2, using Adam optimizer with a fixed learning rate of $10^{-4}$, no weight decay, and gradient clipping at 0.5. A short warm-up phase of one step is used. We use data augmentation including random scaling ($s \sim \mathcal{U}[0.95, 1.05]$), rotation, and vertex jitter with noise level 0.01 and masking.

## E  More Results

### E.1  UV Unfolding

We provide additional qualitative results of SeamGPT on diverse meshes. As shown in Figure 10, our model consistently produces semantically coherent seams that align with structural boundaries.

### E.2  Part Segmentation

Figure 11 shows additional segmentation results enhanced by SeamGPT. The predicted seams help produce cleaner and more precise part boundaries, especially in complex regions.

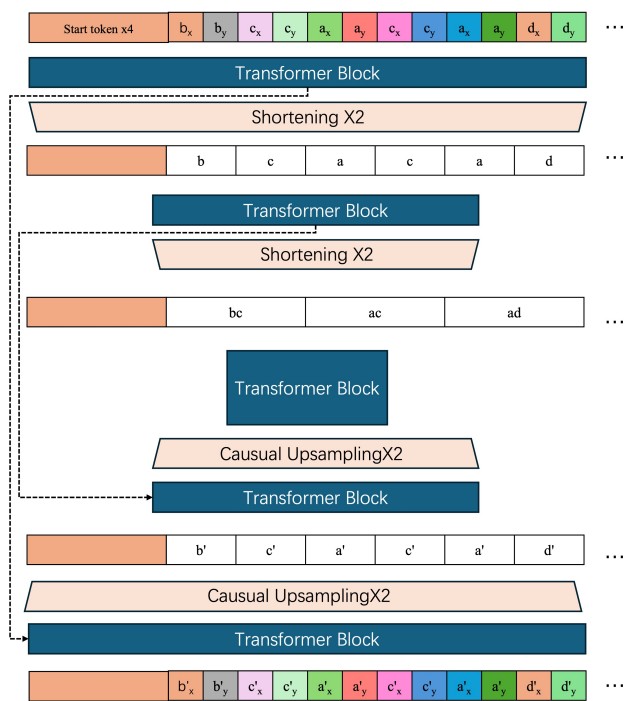

Figure 9: Illustration of SeamGPT decoder architecture under 2D mesh case. SeamGPT uses an Hourglass Transformer backbone similar to Meshtron (Hao et al., 2024b). Tokens at each shortened stage align with the vertices and edges of the seam sequence, providing good inductive bias for seam modeling.

### E.3 AI-GENERATED MESH BENCHMARK

To provide a comprehensive evaluation across different mesh types, we introduce additional quantitative results on a new benchmark consisting of AI-generated meshes. Tables 7 and 8 present the results on this AI-generated mesh benchmark. Our method consistently outperforms XAtlas across all evaluated models, achieving lower average distortion of 16.07 versus 16.63 while significantly reducing UV fragmentation to 30.8 versus 135 fragments on average. Notably, SeamGPT produces substantially fewer UV islands, with a 4.4 reduction in fragment count compared to XAtlas, which is crucial for practical texture editing workflows.

Table 7: Distortion comparison on AI-generated mesh benchmark.

| Method | Average | Whale | Eel | GoldFish | Ordinary | Shark | Rocky | Crocodile | Bird | Elite | Dog |
|---|---|---|---|---|---|---|---|---|---|---|---|
| XAtlas | 16.63 | 4.57 | 16.89 | 36.81 | 38.67 | 21.03 | 10.13 | 4.56 | 8.26 | 18.41 | 6.95 |
| SeamGPT | **16.07** | 6.51 | 7.09 | 37.57 | 38.93 | 10.27 | 11.95 | 7.00 | 27.83 | 7.73 | 5.75 |

Table 8: Fragment count comparison on AI-generated mesh benchmark.

| Method | Average | Whale | Eel | GoldFish | Ordinary | Shark | Rocky | Crocodile | Bird | Elite | Dog |
|---|---|---|---|---|---|---|---|---|---|---|---|
| XAtlas | 135 | 106 | 34 | 44 | 175 | 112 | 141 | 158 | 151 | 290 | 139 |
| SeamGPT | **30.8** | 31 | 7 | 9 | 58 | 31 | 29 | 32 | 35 | 63 | 13 |

These results demonstrate that SeamGPT's learned cutting strategies generalize effectively across diverse mesh generation paradigms, from traditional artist workflows to modern AI-based synthesis methods. The consistent performance improvements across all three benchmarks validate the robustness of our auto-regressive approach for semantically meaningful surface cutting.

## E.4 ABLATION STUDY OF POINT SAMPLING STRATEGY

Table 9 and Table ?? show the sampling results under different edge vertex point ratio and different sampling strategy. Edge:Vert=5:5 got good, balanced results between distortion and fragments. Drastically increasing the Edge ratio or Vert ratio (e.g., 10:0 or 0:10) leads to degraded results. Mixing random points with edge and vert points slightly changes the results. Using Random sampling alone worsens the metrics.

Table 9: Ablation of Edge Vert Sampling ratio.

| Vert: Edge Ratio | 10:0 | 9:1 | 8:2 | 7:3 | 6:4 | 5:5 | 4:6 | 3:7 | 2:8 | 1:9 | 0:10 |
|---|---|---|---|---|---|---|---|---|---|---|---|
| Distortion | 2.343 | 1.922 | 1.912 | **1.903** | 2.064 | 1.907 | 2.419 | 2.927 | 1.922 | 2.066 | 2.258 |
| Fragments | 30.5 | 23.5 | 23.3 | 26.9 | 25.9 | 18.0 | **14.4** | 26.4 | 25.0 | 41.5 | 34.8 |

Table 10: Ablation of Edge Vert Sampling strategy.

| | Vert: Edge= 1:1 | Random | Random : Vert : Edge = 1:1:1 |
|---|---|---|---|
| Distortion | **1.907** | 2.264 | 1.921 |
| Fragments | 18.0 | 32.9 | **16.6** |

## E.5 FRAGMENTS REDUCTION

A critical advantage of SeamGPT is its ability to generate semantically coherent cuts that result in significantly fewer UV fragments compared to traditional methods. Table 11 demonstrates that SeamGPT achieves substantial fragment reduction compared to XAtlas across a comprehensive set of test meshes. The reduction in fragmentation is particularly evident in complex models where maintaining UV coherence is challenging.

Table 11: Fragment count comparison between SeamGPT and XAtlas.

| Method | Avg. | fand. | horse | bimba | lucy | ogre | armad. | bunny | nefert. | dragon | mplk | homer | teapot | happy | che. | spot | all | igea | rocker. | beast | cow |
|---|---|---|---|---|---|---|---|---|---|---|---|---|---|---|---|---|---|---|---|---|---|
| XAtlas | 1292.3 | 14 | 4539 | 7899 | 732 | 635 | 318 | 1045 | 473 | 1652 | 38 | 123 | 50 | 4649 | 66 | 75 | 1 | 2194 | 42 | 1209 | 92 |
| SeamGPT | **195.5** | 14 | 12 | 44 | 627 | 821 | 331 | 26 | 153 | 461 | 4 | 32 | 8 | 161 | 84 | 13 | 6 | 9 | 32 | 1067 | 13 |

The substantial reduction in fragment count has direct practical implications for texture editing workflows. Fewer fragments mean artists can work with more cohesive UV layouts, reducing the complexity of texture painting and improving the visual continuity of applied materials. This advantage is particularly pronounced in production environments where efficient texture workflows are essential for meeting project deadlines and maintaining artistic quality.

## E.6 RUN-TIME

Runtime efficiency is crucial for practical deployment in production pipelines. Table 12 presents a comprehensive runtime comparison across different methods. SeamGPT achieves an average runtime of 35.5 seconds, outperforming XAtlas and demonstrating significant efficiency gains over other learning-based approaches. Our method produces 220 tokens per second on an H20 GPU, enabling real-time interactive workflows.

Table 1 is for the toy4K benchmark, which contains low-poly meshes (average 19K faces). Table 12 is for Flatten-Anything benchmark (average 69K faces). When the number of faces is high, XAtlas's runtime increases significantly, while SeamGPT's runtime remains relatively stable.

SeamGPT demonstrates comparable performance to XAtlas while being two orders of magnitude faster than other learning-based approaches such as Nuvo and FAM. This efficiency advantage stems from our autoregressive generation strategy, which avoids the computationally expensive optimization procedures required by traditional learning-based methods. The combination of high-quality results and fast inference makes SeamGPT particularly suitable for interactive applications and large-scale processing workflows.

## E.7 COMPARISON WITH ARTIST-CREATED UV MAPS

To evaluate the quality of SeamGPT's cutting strategies against human expertise, we conducted a comparison with artist-created UV maps. For fair comparison, professional artists were asked

Table 12: Runtime comparison across different UV parameterization methods.

| Method | **Average** | dragon | igea | happy | bimba |
|--------|---------|--------|------|-------|-------|
| XAtlas | 80.4 | 102 | 61 | 36 | 122 |
| Nuvo | 2925.8 | 3148 | 3255 | 2284 | 3015 |
| FAM | 5656.3 | 6600 | 5302 | 5364 | 5359 |
| SeamGPT | **35.5** | 42 | 27 | 36 | 37 |

to draw surface cuts only, while the flattening process remained identical to SeamGPT's pipeline. This ensures that differences in results stem purely from cutting strategy rather than optimization variations.

Tables 13 and 14 present the comparative results. SeamGPT achieves superior distortion performance with an average of 16.07 compared to artists' 18.61, demonstrating that our learned approach can identify cutting strategies that outperform human intuition in terms of geometric quality. The fragment count comparison shows that SeamGPT produces slightly more fragments on average (30.8 vs. 26.3), representing a reasonable trade-off for the improved distortion characteristics.

Table 13: Distortion comparison between SeamGPT and artist-created cuts.

| Method | **Average** | Whale | Eel | GoldFish | Ordinary | Shark | Rocky | Crocodile | Bird | Elite | Dog |
|--------|---------|-------|-----|----------|----------|-------|-------|-----------|------|-------|-----|
| Artists | 18.61 | 6.83 | 15.28 | 44.18 | 55.94 | 20.62 | 8.93 | 2.99 | 7.70 | 18.18 | 5.44 |
| SeamGPT | **16.07** | 6.51 | 7.09 | 37.57 | 38.93 | 10.27 | 11.95 | 7.00 | 27.83 | 7.73 | 5.75 |

Table 14: Fragment count comparison between SeamGPT and artist-created cuts.

| Method | **Average** | Whale | Eel | GoldFish | Ordinary | Shark | Rocky | Crocodile | Bird | Elite | Dog |
|--------|---------|-------|-----|----------|----------|-------|-------|-----------|------|-------|-----|
| Artists | **26.3** | 29 | 10 | 9 | 36 | 33 | 21 | 27 | 32 | 39 | 27 |
| SeamGPT | 30.8 | 31 | 7 | 9 | 58 | 31 | 29 | 32 | 35 | 63 | 13 |

These results highlight SeamGPT's ability to discover cutting patterns that balance geometric fidelity with topological simplicity. While artists naturally prioritize semantic boundaries and aesthetic considerations, SeamGPT's data-driven approach optimizes for measurable quality metrics, resulting in lower overall distortion at the cost of slightly increased fragmentation.

### E.8 FURTHER DISCUSSION.

*Tokenization Process.* The current tokenization process follows Meshtron (Hao et al., 2024b). In our early experiment, we also implemented the BPT (Weng et al., 2024)-style tokenization process for the SeamGPT. But the training and inference are less stable than the current implementation.

*Curvature feature.* When implementing the Edge-CLS baseline (c.f. line 314), we concatenate point normal, xyz, with curvature as the input feature to the graph neural network. The Edge-CLS model then classifies the edge into seam/non-seam with a mesh graph neural network backbone. The curvature was computed using trimesh's built-in function. The problem is that the training data contains a large portion of non-manifold meshes (the data is largely obtained from the objaverse dataset which are largely manmade meshes). which makes the curvature computation unstable and the resulting curvature feature noisy. In addition, computing curvature could be time-consuming when the mesh has large number of polygons. The results on Edge-CLS experiment do not show much difference between the model with and without curvature feature.

*Partial Shape.* The method can work for partial shapes. This is because the original training data contains many partial shapes, such as human body parts like single arms or legs. In our later experiments, we added more partial data for augmentation. This significantly improved the model's robustness to partial shapes.

*Diffusion Baseline.* Diffusion for seam prediction is an interesting idea. However, the length of seams varies significantly, ranging from tens of tokens to tens of thousands of tokens. Since diffusion models require fixed-length inputs, we would need to pad tokens to a maximum length limit for training and inference. Diffusion models are less effective at handling variable-length sequences, and the maximum length of predicted seams would be limited. For reference, diffusion-based mesh generation works such as MeshCraft (He et al., 2025) use flow-based diffusion models to generate meshes. MeshCraft works only on simple low-poly meshes and fails to generate complex high-poly

meshes. The performance of MeshCraft is not as good as AR-based models such as BPT (Weng et al., 2024) and Meshtron (Hao et al., 2024b).

*Rotation Handling.* During training, we randomly rotate and apply uniform scaling [0.8 1.0] to the mesh before tokenization. After rotation and scaling, the vertex coordinates become different words. For autoregressive 3D generalization, we are less concerned with rotation invariance. Instead, it is more important for the model to learn as many words as possible. Random rotation and scaling can significantly enrich the vocabulary of the autoregressive model, thereby improving its robustness.

*Does large triangles pose a problem?* We do not handle large triangles specially. We apply the same sampling strategy to all triangles. Large triangles do not pose any problem: as shown in table 15. The Toy4K benchmark contains the largest triangles, but it gets the lowest distortion. We can not observe any significant correlations between the triangle size and the UV flattening results.

Table 15: Results on benchmark with different triangle size.

| Benchmark | Toy4K | FAM | AI-Generated |
|---|---|---|---|
| Avg. triangle size (normalized) | 3.92e-3 | 6.95e-4 | 1.20e-3 |
| Distortion metric | **1.95** | 13.04 | 16.7 |
| Fragment count metric | 153 | 195.5 | **30.8** |

*Is distortion an indicator of semantic superiority?* Low distortion does not indicates semantic superiority. Instead, good seams are balanced between semantics and distortion. These two things (1) Low distortion (2) Semantically alignment are complementary and cannot be evaluated separately. They may be mutually exclusive. One can achieve zero distortion UV mapping by cutting all triangles into separate UV fragments, i.e., over-cutting is a simple way to achieve low distortion. However, this would result in a very large number of UV fragments, making texture creation and editing very difficult. Therefore, good seams balance between semantics and distortion.

*Why seams should align with semantic parts?* 3D artist prefer to cut along semantic boundaries to obtain semantically coherent UV islands. This makes assigning materials to each part much easier and also make texture editing easier. SeamGPT is data-driven; the training seams are artist-created. SeamGPT therefore learns to cut along semantic boundaries.

*Quantization resolution.* Increasing quantization resolution leads to more precise seam coordinates. However, the vocabulary size also increases, which requires more training data and makes training more difficult. In our early experiments, we found that a quantization resolution of 2048 makes training unstable, and 512 makes the seam too coarse. A quantization resolution of 1024 balances seam precision with training/inference stability.

*Seam-length control.* The seam length condition can control the approximate length of seams, but cannot guarantee the exact length of seams. Since the shape condition is also injected into the AR decoder, the final length also depends on the complexity of the specific shape.

*Drawback of UV.* The drawbacks of UVs mainly arise from automatic UV unwrapping, as indicated in the cited paper (Foti et al., 2024). However, our training data consists of artistically created UVs with professionally designed cuts, which are clean and less prone to flaws. We also adopted a rigorous data filtering process to remove UVs with artifacts (see line 179), such as data from scanned meshes that tend to rely on auto-UV tools.

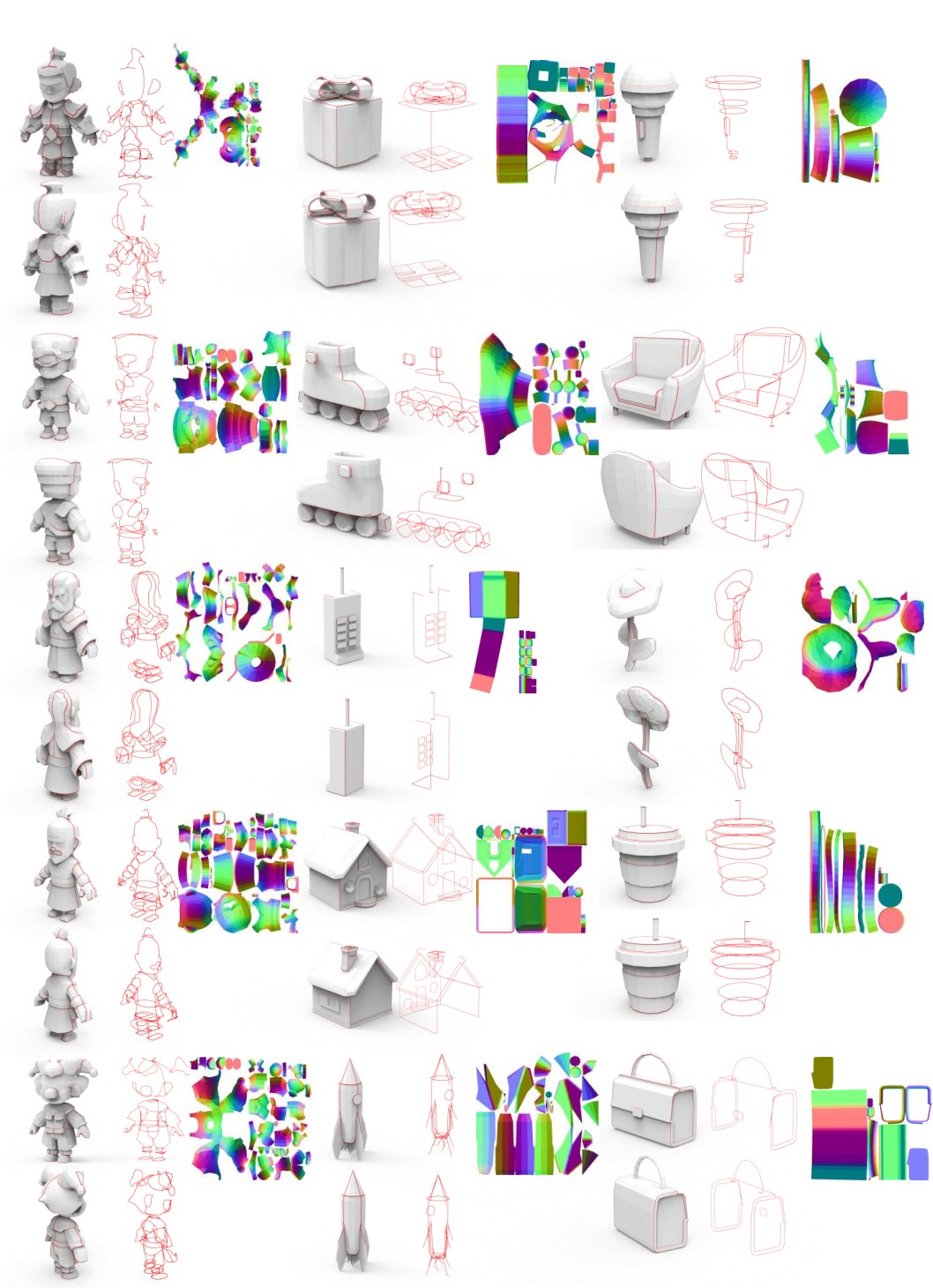

Figure 10: Seam prediction and UV unwrapping results. The left column displays the mesh with predicted seams overlaid, the middle column shows the predicted seam lines, and the right column presents the corresponding UV unwrapping results.

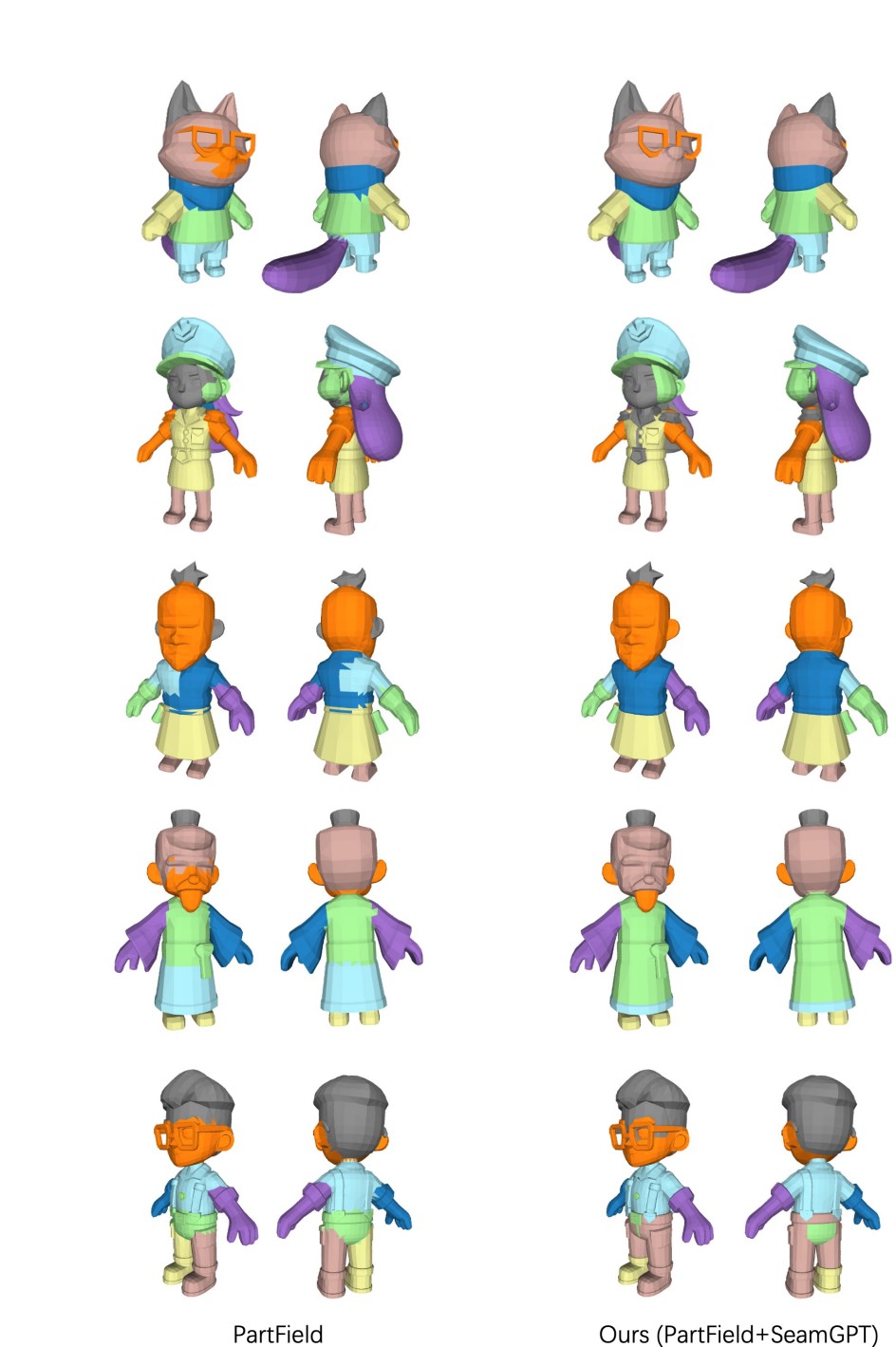

PartField        Ours (PartField+SeamGPT)

Figure 11: More part segmentation results. Last row shows the **failure case**: When SeamGPT fails to cut along the semantic boundary (e.g., between the leg and hips), this results in inconsistent segmentation with the semantics provided by PartField.

