# OpenReview forum: "Auto-Regressive Surface Cutting"
_ICLR.cc/2026/Conference — Submitted to ICLR 2026_

### Official Review · Reviewer_4sRE · 2025-10-27

**Soundness:** 3
**Presentation:** 3
**Contribution:** 3
**Rating:** 6
**Confidence:** 4

**Summary:**

This paper proposes SeamGPT, an auto-regressive transformer model for generating surface-cutting seams on 3D meshes.
The authors reformulate surface cutting — a classic problem in UV parameterization and mesh segmentation — as a sequence generation task, treating each 3D coordinate of a seam segment as a discrete token predicted autoregressively.
A point-cloud encoder encodes sampled vertices and edges into a latent shape embedding, which conditions a GPT-style hourglass transformer decoder to sequentially predict seam coordinates.
The method is trained on a large dataset (≈560K artist-annotated meshes filtered from Objaverse/3D-Future) and evaluated on UV unwrapping and part segmentation benchmarks.

While the paper presents a polished system and thorough experiments, the core technical novelty is minimal. Most components (point-cloud encoder, quantization, autoregressive transformer, hourglass hierarchy) are directly adopted from existing works such as MeshGPT (Siddiqui et al. 2023) and Meshtron (Hao et al. 2024) with only superficial adaptation to a new task.
As a result, the overall contribution feels incremental and more of an application of existing sequence modeling methods than a fundamental advance in 3D geometry understanding.

**Strengths:**

The paper’s strength lies mainly in its problem framing rather than architectural innovation. It takes an established operation in computer graphics—surface cutting and UV unwrapping—and recasts it as an auto-regressive generation task. This conceptual reformulation is novel from a modeling perspective and has some intuitive appeal, as it parallels how artists progressively define seams in practice. The proposed workflow is well structured and technically sound: it integrates a point-cloud encoder to extract shape features and a transformer decoder for coordinate generation, both of which are established and well-tested components. The experimental section is extensive, evaluating across multiple benchmarks (FAM, Toys4K, AI-generated meshes), with both quantitative metrics and qualitative visualizations. The results consistently show that SeamGPT can produce fewer fragmented UV charts with comparable or slightly lower distortion compared to baseline methods. The inclusion of user studies with professional artists is also commendable, as it adds a perceptual validation dimension to the otherwise geometric evaluations. The paper is generally well written and easy to follow, and the authors provide sufficient implementation details for reproduction.

**Weaknesses:**

Despite a polished presentation, the paper’s technical novelty and empirical contributions are quite limited. The proposed framework is almost entirely composed of pre-existing components: the point-cloud encoder is borrowed from standard 3D transformer models, and the auto-regressive decoder closely follows prior work such as PolyGen, MeshGPT, and related sequence models for 3D geometry. The only distinct aspect—the reformulation of cutting seams as quantized coordinate sequences—feels incremental rather than conceptually new.

More concerning, the experimental results do not convincingly support the claimed superiority. In Tables 2 and 3, SeamGPT’s distortion metrics are often worse than or comparable to XAtlas, which is a purely geometric non-learning baseline. For example, on several FAM models (e.g., Bimba, Dragon, Happy Buddha) and most Toys4K categories, XAtlas yields lower distortion. The authors highlight fragment reduction, but this is a side effect of predicting fewer seams rather than genuine geometric improvement. The evaluation lacks a rigorous statistical analysis or significance testing, and the results fluctuate widely across categories.

**Questions:**

- Why does SeamGPT perform worse than Xatalas in distortion metrics on many test cases if its purpose is to improve UV quality?
- How sensitive are the results to the quantization resolution and seam-length control parameter?

---

> ### Author Response · Authors · 2025-11-19
> **Official Comment by Authors**
>
> We thank the reviewer for the constructive comments. We provide our answers below and also upload the revised paper.
>
>
> ### **[W1] Novelty**
> We appreciate the reviewer's concern about novelty. However, we believe our contribution is significant for the following reasons:
> (1)While the components may be familiar, reformulating UV unwrapping as a sequence generation problem is conceptually novel and enables the first deep learning solution to this long-standing problem.
> (2)Our method achieves substantial improvements (85\% fewer fragments with lower distortion than XAtlas), demonstrating that the proposed formulation effectively addresses the core challenges of UV unwrapping.
> (3)The ability to balance distortion and fragment count is crucial for real-world applications, where our method provides a significant advantage over existing approaches.
>
> ### **[W2] XAtlas gets good distortion in many cases.**
> Distortion is not the only metric for evaluating UV quality. **One can achieve zero distortion UV mapping by cutting all triangles into separate UV fragments, i.e, over-cutting is a simple way to achieve low distortion**.
> However, this would result in a very large number of UV fragments, making texture creation and editing very difficult.
> XAtlas achieves low distortion by over-cutting. SeamGPT achieves comparable distortion with much fewer cuts.
> As shown in the table below (evaluation result on FAM benchmark), our method manages to achieve lower distortion with 85\% fewer UV fragments than XAtlas.
>
> | Method | Distortion | # Fragments|
> |--------|------------|-----------------------------------|
> | XAtlas | 18.37| 1292.3|
> | Ours| **13.04** | **195.5** |
>
>
>
> ### **[Q1] Quantization resolution**
> Increasing quantization resolution leads to more precise seam coordinates.
> However, the vocabulary size also increases, which requires more training data and makes training more difficult.
> In our early experiments, we found that a quantization resolution of 2048 makes training unstable, and 512 makes the seam too coarse.
> A quantization resolution of 1024 balances seam precision with training/inference stability.
>
>
> ### **[Q2] Seam-length control**
> The seam length condition can control the approximate length of seams, but cannot guarantee the exact length of seams.
> Since the shape condition is also injected into the AR decoder, the final length also depends on the complexity of the specific shape.

---

### Official Review · Reviewer_5xei · 2025-10-30

**Soundness:** 3
**Presentation:** 3
**Contribution:** 2
**Rating:** 4
**Confidence:** 3

**Summary:**

This paper introduces a GPT-based auto-regressive model to infer and generate cutting seams derived from data. The surface cutting problem is turned into a next token prediction task and a language-like learning is conducted to infer seams. The experiments on several meshes demonstrate the seams inferred this way can better align artists' preferences.

**Strengths:**

- Overall, it's a good idea and a low hanging fruit to approach seam cutting through a GPT-like architecture.
- I believe that the results are accurate and good quality can be achieved this way.
- The paper clearly describes the approach and except some doubts on implementation (see below) the work seems to be reproducible.

**Weaknesses:**

- Abstract claims exceptional performance. This is not validated by the experiments. Please tone down.
- Missing baselines (See Q1).
- UV texturing can introduce bad seams. There are some new works that discuss and alleviate this problem. See:
Foti, S., Zafeiriou, S., & Birdal, T. Uv-free texture generation with denoising and geodesic heat diffusion. NeurIPS 2024. More on this in Q3 below.
- Ordering (for example yzx) is rotation dependent and there seems to be no treatment of this. See Q4.
- Large triangles seem to be problematic for this work. Point cloud networks that operate on mesh vertices will fail if the surface is not resampled. Large triangles will also cause more errors in the seams which are directly defined over edges / vertices. See Q6.
- In principle, edges have infinite number of points whereas vertices are finite. Even sampling on both does not seem justified. This must be studied in a controlled manner. See Q7.
- I'm not sure if qualitative examples in Fig. 3 are conclusive enough to justify the 'artistic' quality superiority.
- Appendix seem to report that SeamGPT is faster in runtime, whereas table 1 shows that it's slower. There seems to be some inconsistency or not consistent benchmarking of runtime.
- Paper uses graph convolutions as baselines for mesh processing. This is not okay. I suggest comparing to any mesh convolution based network.
- Before making conclusive statements, I would like to see the results before Blender's minimum stretch algorithm is applied. One needs to gauge how much of the actual contribution is coming from this. In fact, this should be applied to other methods as well, for a fair comparison.
- Social impact: This work is not theoretical. It has immediate practical application and can be used by artists. As such, I invite the authors to think a little about the implications of their work rather than dismissing this mandatory section. We owe this much to our community.

Minor weaknesses:
- Ln. 41: flatten -> flattening
- Ln. 64: init -> initialize
- Ln. 203: S has to be ordered, not a set. This is true for all in Eq. 1. In fact I'm not sure if Eq. 1 is actually needed. It is covered in preceding paragraph.
- Ln. 267: Isn't H20 96GB? (text says 98)
- Ln. 367: S was reserved for sequence, now used for depicting a 3D shape
- "Does pointer networks work?" section is not an ablation study. Please use the term correctly.

**Questions:**

1. What about a non auto-regressive method based on for example diffusions? Can we make a simple baseline and compare?
2. Will the authors make the filtered dataset public? (Maybe indices of the models?)
3. UV-parameterization naturally suffers from introducing arbtirary seams that are not meant to be in the original shape. So what about these seams that had to be there not because of semantics but because of the drawbacks of UV?
4. How are rotations of the meshes handled? An equivariant network? It feels like data augmentation would just cause additional problems here.
5. How is the quantization in Ln. 219 precisely done?
6. What about large triangles? How are those handled? Are there different resampling strategies? Any of these ablated?
7. Why are the points on vertices and edges evenly split? What about other ratios?
8. Does the paper compare to MeshGPT encoder?
9. How about using a test set that is split from the training set in the experiments? Did the authors try this?
10. In experiments, why is distortion an indicator of semantic superiority?
11. How is the seam lines are used to partition the shape into P_i as in Ln. 374?
12. I don't see any reason why seams should align with semantic 3D parts of the objects. Could this be justified?
13. Can we have quantitative results corresponding to Fig. 5 and maybe compare with some other sampling methods?
14. How is R chosen in practice? I mean the actual value.

---

> ### Author Response · Authors · 2025-11-19
> **Official Comment by Authors (part1)**
>
> We thank the reviewer for the constructive comments. We provide our answers below and also upload the revised paper.
>
> ### **[W1] Abstract claims exceptional performance.**
> We fixed the abstract claims.
>
>
> ### **[W2] & [Q1] Diffusion-based baseline**
> Diffusion for seam prediction is an interesting idea.
> However, the length of seams varies significantly, ranging from tens of tokens to tens of thousands of tokens.
> Since diffusion models require fixed-length inputs, we would need to pad tokens to a maximum length limit for training and inference.
> Diffusion models are less effective at handling variable-length sequences, and the maximum length of predicted seams would be limited.
> For reference, diffusion-based mesh generation works such as MeshCraft  use flow-based diffusion models to generate meshes.
> MeshCraft works only on simple low-poly meshes and fails to generate complex high-poly meshes.
> The performance of MeshCraft is not as good as AR-based models such as BPT[Weng et al., 2024] and Meshtron[Hao et al., 2024].
>
> [MeshCraft] He et al., 2025. MeshCraft: Exploring Efficient and Controllable Mesh Generation with Flow-based DiTs
>
>
>
>
> ### **[W3] Noisy UV cuts due to drawbacks of UV**
> If some seams are not semantically necessary but are added to adapt to UV constraints, we consider these cuts valid rather than noise.
> Note that our ultimate goal is that seam generation serves UV unwrapping, with certain tolerance of semantically meaningless cuts.
>
>
> ### **[W4] & [Q4] Rotation handling**
> During training, we randomly rotate and apply uniform scaling [0.8~1.0] to the mesh before tokenization.
> After rotation and scaling, the vertex coordinates become different words.
> For autoregressive 3D generalization, we are less concerned with rotation invariance.
> Instead, it is more important for the model to learn as many words as possible.
> Random rotation and scaling can significantly enrich the vocabulary of the autoregressive model, thereby improving its robustness.
>
>
> ### **[W5] & [Q6] Large triangles**
> We do not handle large triangles specially. We apply the same sampling strategy to all triangles.
> Large triangles do not pose any problem: as shown in the table below, the Toy4K benchmark contains the largest triangles, but it gets the lowest distortion.
> We can not observe any significant correlations between the triangle size and the UV flattening results.
>
> | Benchmark | Toy4K | FAM | AI-Generated |
> |--------|-------|-----|-------------|
> | Avg. triangle size (normalized) | 3.92e-3 | 6.95e-4 | 1.20e-3 |
> | Distortion metric | **1.95** | 13.04 | 16.7 |
> | Fragment count metric | 153 | 195.5 | **30.8** |
>
>
> ### **[W6] & [Q7] & [Q13] Ablation study of Sampling Strategy**
>
> The following 2 tables show the sampling results under different edge vertex point ratio and different sampling strategy.
> Edge:Vert=5:5 got good, balanced results between distortion and fragments. Drastically increasing the Edge ratio or Vert ratio (e.g., 10:0 or 0:10) leads to degraded results.
> Mixing random points with edge and vert points slightly changes the results.
> Using Random sampling alone worsens the metrics. We have added this in the revised paper.
>
> |  Vert: Edge Ratio | 10:0 | 9:1 | 8:2 | 7:3 | 6:4 | 5:5 | 4:6 | 3:7 | 2:8 | 1:9 | 0:10 |
> |----------------|------|-----|-----|-----|-----|-----|-----|-----|-----|-----|------|
> | Distortion  | 2.343 | 1.922 | 1.912 | **1.903** | 2.064 | 1.907 | 2.419 | 2.927 | 1.922 | 2.066 | 2.258 |
> | Fragments  | 30.5 | 23.5 | 23.3 | 26.9 | 25.9 | 18.0 | **14.4** | 26.4 | 25.0 | 41.5 | 34.8 |
>
>
>
>
>
> | Strategy | Distortion | Fragments |
> |------|------|------|
> | Vert: Edge= 1:1 | **1.907** | 18.0 |
> | Random | 2.264 | 32.9 |
> | Random : Vert : Edge = 1:1:1 | 1.921 | **16.6** |
>
>
>
> ### **[W7] justify the 'artistic' quality superiority**
> The examples shown in Figure 3 demonstrate that SeamGPT's unwrapping results are more semantically meaningful.
> For instance, in the cow example, the cow's body is split along the midline, and the legs and horns are cut reasonably. Compared to other baselines, this better aligns with semantic boudaries.
> The user study also confirms this. **Note that the user study involved 20 professional 3D artists.**
>
>
> ###  **[W8] Runtime inconsistency**
> The runtimes are different, because they are on two different benchmarks.
> Table 1 is for the toy4K benchmark, which contains low-poly meshes (average 19K faces).
> Table 9 in the Appendix are for Flatten-Anything benchmark (average 69K faces).
> When the number of faces is high, XAtlas's runtime increases significantly, while SeamGPT's runtime remains relatively stable.
>
> ### **[W9] Mesh Baseline**
> We use an 8-layer SAGEConv followed by 21-layer full attention transformer layers as our mesh baseline. We believe this configuration is strong enough to reflect the potential of the mesh baseline.

---

> > ### Author Response · Authors · 2025-11-19
> > **Official Comment by Authors (part2)**
> >
> > ### **[W10] Results before Blender's minimum stretch algorithm**
> > The baselines XAtlas, Nuvo and FAM are single-stage algorithms: they directly output the UV-Flatten result. Thus, it is not possible to apply minimum stretch algorithm to XAtlas, Nuvo and FAM baselines.
> > Edge-CLS and Ours are two-stage algorithms: (1) predicting seams on the surface first, and (2) flattening with Blender's minimum stretch.
> > Therefore, **the results before Blender's minimum stretch are the predicted 3D seams, which are already shown in Figure 3.**
> > Before Blender's minimum stretch, SeamGPT produces more reasonable seams than Edge-CLS.
> > Blender's minimum stretch alone, without any cutting seams, can also flatten the shape (i.e. flattening the entire shape into a single UV island), this will cause tremendous distortion.
> > We have clarified this in Figure 3.
> >
> >
> > ### **[W11] Social impact**
> > Our paper presents SeamGPT, a learned approach for UV unwrapping and surface cutting.
> > It establishes the foundation for an important research direction in automated 3D geometry processing,
> > which is needed for a variety of applications where high-quality 3D content creation and manipulation is required.
> > These applications range from the fields of game development and digital entertainment to architectural visualization and industrial design.
> > In the former, efficient and high-quality UV unwrapping is of major importance in order to enable realistic texture mapping and material editing.
> > Applications such as virtual world construction and 3D asset generation would greatly benefit from research like ours.
> > This, in turn, could provide society with more accessible tools for 3D content creation, lowering the barrier for artists and designers.
> > On the other hand, as a low-level building block for 3D geometry processing, our work has no direct negative outcome,
> > other than what could arise from the aforementioned applications. We have added this in the revised paper.
> >
> >
> > ### **[W12] Minor weaknesses**
> > Thanks, we fixed all minor weaknesses and updated the paper.
> >
> >
> >
> >
> >
> > ### **[Q2] Dataset Release**
> > We will release the ID of training models in Objaverse/ObjaverseXL upon paper acceptance.
> >
> >
> >
> >
> > ### **[Q5] How is the quantization done?**
> > Quantization is done by rounding floating-point coordinates to the nearest integer. For example, (127.23, 933.56, 101.10) → (127, 934, 101).
> >
> >
> >
> >
> > ### **[Q8] Compare with MeshGPT's encoder**
> > We actually already compared with a baseline that is similar to MeshGPT's encoder in the paper.
> > MeshGPT use a Mesh-level encoder which contains 5 layers of SAGEConv, to encode the mesh into a latent vector.
> > We implemented a mesh encoder similar to MeshGPT's encoder as a strong baseline.
> > Our mesh encoder contains 8 layers SAGEConv followed by 21 full self-attention blocks, i.e., the total number of parameters is much larger than MeshGPT's. We clarified this in the revised paper.
> > The result can be seen in Figure 6. The mesh encoder is much worse than the point cloud encoder.
> >
> >
> > ### **[Q9] Training/validation set split**
> > During training, we use the validation set to evaluate the performance of the model.
> > The validation set consists of 2K objects, randomly sampled from the training set.
> > This was mainly to observe the training progress and avoid overfitting. The validation loss decreases consistently with the training loss.
> >
> >
> >
> >
> >
> >
> > ### **[Q10] In experiments, why is distortion an indicator of semantic superiority?**
> > Note that we did not say that low distortion indicates semantic superiority. Instead, good seams are balanced between semantics and distortion (c.f. line 61).
> > These two things (1) Low distortion (2) Semantically alignment are complementary and cannot be evaluated separately.
> > They may be mutually exclusive. One can achieve zero distortion UV mapping by cutting all triangles into separate UV fragments, i.e., over-cutting is a simple way to achieve low distortion**.
> > However, this would result in a very large number of UV fragments, making texture creation and editing very difficult.
> > Therefore, good seams balance between semantics and distortion.
> >
> >
> >
> > ### **[Q11] Shape partition method**
> > After surface cutting, shape partition is done by clustering connected faces. We have clarified this in the revised paper.
> >
> >
> > ### **[Q12] Why seams should align with semantic parts?**
> > 3D artist prefer to cut along semantic boundaries to obtain semantically coherent UV islands. This makes assigning materials to each part much easier and also make  texture editing easier.
> > SeamGPT is data-driven; the training seams are artist-created. SeamGPT therefore learns to cut along semantic boundaries.
> >
> >
> >
> >
> > ### **[Q14] How is R chosen in practice?**
> > As shown in Table 5. increasing R lead to lower distortion but more fragmented UV map.
> > R is empirically set to 0.2 to balance between distortion and fragment count. We have clarified this in the revised paper.

---

> > > ### Comment · Reviewer_5xei · 2025-11-24
> > >
> > > Given that:
> > > (i) direct generation is applied in language processing, so variants of diffusion modeling do handle variable length inputs;
> > > (ii) the paper is extra defensive on the benefits of UV and would not admit the drawbacks of this as a representation;
> > > (iii) comparisons with FAM benchmark being seemingly disadvantageous for FAM;
> > > (iv) the revised version did not include some of my suggestions or some of the discussion mentioned, and does not cite the literature;
> > >
> > > I prefer to keep my original rating. Additionally, while subjective, simply extending autoregressive modeling to seam generation does not provide me an interesting insight.

---

> > > > ### Author Response · Authors · 2025-11-24
> > > >
> > > > ### **Direct generation**
> > > > We are aware of existing work on Diffusion for LLMs. DiffusionLLM has certain advantages: for example, it already utilizes bidirectional full attention. However, in the domain of large language models, the performance of DiffusionLLM still falls short of mainstream autoregressive (AR) models.
> > > >
> > > > But we are very willing to dig deeper into the Diffusion approach for SEAM prediction as a standalone project (e.g., developing an efficient seam compression network to compress it into a latent space, and generate it with a Dit ). We plan to incorporate this into our future work.
> > > >
> > > >
> > > > ### **Drawback of UV.**
> > > > The brawback of UV are mainly arised from automatic UV un-wrapping, as shown in the raise paper (Foti, S., Zafeiriou, S., & Birdal, T. Uv-free texture generation with denoising and geodesic heat diffusion. NeurIPS 2024).
> > > > Our training data mainly consists of artistically created UVs with professionally designed cuts, which are clean and less prone to flaws.
> > > > We also adopted a rigorous data filtering process to remove UVs with artifacts (c.f. line 179), such as data from scaned mesh (which tend to rely on a auto-uv tool).
> > > > We discussed this in the revised paper and added the relevant citations.
> > > >
> > > >
> > > >
> > > > ### **Metric discrepancy**
> > > > The metric in FAM, and this paper are actually different. See the reply at https://openreview.net/forum?id=9HeKCYl1zl&noteId=qhZ1SV2LC4
> > > >
> > > > ### **Paper updata**
> > > > We upload a revised version to reflect all disscussion and suggestions.
> > > >
> > > >
> > > > ### **Insights**
> > > > UV unwrapping is an important task in traditional graphics pipeline. We are the first to tackle this problem using an autoregressive method and achieve good results. We believe our work provides insights for researchers working on this actual problem.

---

### Official Review · Reviewer_nRhE · 2025-10-31

**Soundness:** 4
**Presentation:** 3
**Contribution:** 4
**Rating:** 10
**Confidence:** 5

**Summary:**

The paper formulates surface cutting as a next token prediction task, and designs a novel auto-regressive architecture that predicts seam coordinate sequences from a given mesh.

**Strengths:**

1. The paper proposes a new paradigm for surface cutting, which formulates surface cutting as a next token prediction task. The idea is novel, and the results are pretty good.
2. Surface cutting is a very important task in 3D understanding. It essentially finds the best way to geometrically segment a 3D surface into parts (with different criteria). With the part information, it potentially boosts a variety of downstream tasks, such as semantic segmentation (as demonstrated), texture editing, rendering, generation, animation, articulated objects, etc. Hence, the contribution of this paper to the community is significant in my opinion.

**Weaknesses:**

1. As the method is trained purely supervised by ground truth cuttings, the quality of the ground truth cuttings matters a lot, and the model might be sensitive to the poor samples. As the authors mentioned, a rigorous filtering process was applied to clean the data. Thus, scaling up the dataset may be laborious.
2. Some details about the paper are not clearly described, which I will mention in the question section.

**Questions:**

1. How does the number of sampled points affect the performance?
2. How to choose K at line 244?
3. In the data augmentation, how large a portion will the masked region be? Is the method able to predict seams for a part of the object (instead of feeding in the whole point cloud, feed in a point cloud sampled from a part of the object)?
4. The topologies of the objects shown in the paper are fairly simple. How are the generated cuttings for objects with complex topologies? Is this limited by the number of mesh faces?
5. Instead of controlling the segment count by the seam length, is it possible to do this hierarchically? For example, predict basic seams that cut the surface into a small number of segments, and then cut each of the large segments into smaller segments hierarchically.
6. Will the dataset and code be open-sourced?

---

> ### Author Response · Authors · 2025-11-19
> **Official Comment by Authors**
>
> We thank the reviewer for the constructive comments. We provide our answers below and also upload the revised paper.
>
>
>
> ### **[Q1] Number of sampled points**
> We samples 61k points evenly split to edge and vertices.
> We did not observe significant performance drop when using less samples, e.g. 40K and 20K points.
> But changing the sampling ratio between edge and vertice can lead to significant performance drop.
> The following 2 tables show the sampling results under different edge vertex point ratio.
> Edge:Vert=5:5 got good, balanced results between distortion and fragments. Drastically increasing the Edge ratio or Vert ratio (e.g., 10:0 or 0:10) leads to degraded results.
>
> |  Vert: Edge Ratio | 10:0 | 9:1 | 8:2 | 7:3 | 6:4 | 5:5 | 4:6 | 3:7 | 2:8 | 1:9 | 0:10 |
> |----------------|------|-----|-----|-----|-----|-----|-----|-----|-----|-----|------|
> | Distortion  | 2.343 | 1.922 | 1.912 | **1.903** | 2.064 | 1.907 | 2.419 | 2.927 | 1.922 | 2.066 | 2.258 |
> | Fragments  | 30.5 | 23.5 | 23.3 | 26.9 | 25.9 | 18.0 | **14.4** | 26.4 | 25.0 | 41.5 | 34.8 |
>
>
>
> ### **[Q2] How to choose K**
> K is not chosen but computed based on the edge length.
> Since we assign totally 30,720 points to the edges, if an edge's length accounts for a proportion p of the total length of all edges, the number of points allocated to that edge is 30,720 * p.
> we have clarified this in the revised paper.
>
> ### **[Q3] Partial shape**
> The method can work for partial shapes.
> This is because the original training data contains many partial shapes, such as human body parts like single arms or legs.
> In our later experiments, we added more partial data for augmentation.
> This significantly improved the model's robustness to partial shapes.
>
>
>
> ### **[Q4] Complex topologies**
> Complex shapes with a high number of faces are still challenging.
> This makes AR prediction slower (as the seam sequence becomes longer) and unstable.
> Instead of using raw coordinates, we are working on more compact and efficient representation of seam lines.
>
>
> ###  **[Q5] Hierarchical seam generation**
> Hierarchical seam generation is an interesting idea.
> Two key challenges are: (1) how to efficiently compress seam lines, and (2) upsampling from coarse representation to fine representation.
> We are working on this.
>
>
> ### **[Q6] Open source**
> We will open source the code and dataset after the paper is accepted.

---

### Official Review · Reviewer_fdf2 · 2025-11-05

**Soundness:** 3
**Presentation:** 3
**Contribution:** 3
**Rating:** 6
**Confidence:** 4

**Summary:**

This paper introduces SeamGPT, an auto-regressive model for 3D surface cutting and UV unwrapping. The method reformulates surface cutting as a sequence generation problem, where cutting seams are predicted token by token within a quantized 3D space. SeamGPT achieves strong results on both UV parameterization and part segmentation benchmarks.

**Strengths:**

The idea of framing surface cutting as an auto-regressive sequence prediction task is novel and well-motivated. The integration with PartField yields particularly clean and semantically coherent part boundaries, leading to visually impressive segmentation outcomes. The approach also demonstrates solid generalization across datasets and diverse mesh types.

**Weaknesses:**

The method section would benefit from additional toy visualizations to clarify the intuition behind the sequence representation and the quantization/tokenization strategy. Some architectural details (such as hierarchy levels and quantization schemes) could be illustrated more intuitively to help readers grasp the overall process.

**Questions:**

Could the authors add simple illustrative examples (e.g., 2D surfaces or flat cubes) to show how the auto-regressive process operates geometrically and how seams are tokenized?

Since vertex coordinates are quantized and generated sequentially, could BPT-style point compression or token sparsification be integrated into SeamGPT’s decoder to improve efficiency?

During data preparation, did the authors consider incorporating feature-line extraction or curvature-sensitive priors beyond UV seams to better capture subtle geometric cues?

The fandisk example shows missing curved boundaries, would integrating differential geometric features (such as curvature flow) help improve seam placement in such cases?

---

> ### Author Response · Authors · 2025-11-19
> **Official Comment by Authors**
>
> We thank the reviewer for the constructive comments. We provide our answers below and also upload the revised paper.
>
> ### **[W1] & [Q1] Illustrative example for tokenization and decoder architecture.**
> We added 2D examples in Figure 8 and Figure 9 in Appendix to illustrate the tokenization process and the decoder's hierarchical network structure.
> The supplementary video shows how the seams are generated one by one on a 3D avatar example.
>
>
> ### **[Q2] BPT-style tokenization**
> The current tokenization process follows Meshtron (Hao et al. 2024).
> In our early experiment, we also implemented the BPT-style tokenization process for the SeamGPT.
> But the training and inference are less stable than the current implementation.
>
>
> ### **[Q3] & [Q4] Incorporating curvature feature**
> We did try to incorporate the curvature feature in our early experiment.
> When implementing the Edge-CLS baseline (c.f. line 314), we concatenate point normal, xyz, with curvature as the input feature to the graph neural network.
> The Edge-CLS model then classifies the edge into seam/non-seam with a mesh graph neural network backbone.
> The curvature was computed using trimesh's built-in function.
> The problem is that the training data contains a large portion of non-manifold meshes (the data is largely obtained from the objaverse dataset which are largely manmade meshes).
> which makes the curvature computation unstable and the resulting curvature feature noisy.
> In addition, computing curvature could be time-consuming when the mesh has large number of polygons.
> The results on Edge-CLS experiment do not show much difference between the model with and without curvature feature.

---

### Comment · Area_Chair_bwbJ · 2025-11-23
**The reported results of FAM are much worse than those in the original paper**

I have serious concerns regarding the fairness of the comparison with FAM, an unsupervised method. The reported FAM results appear significantly worse than those presented in the original paper. The authors should provide a clear explanation to address this discrepancy.

---

> ### Author Response · Authors · 2025-11-24
> **Metric discrepancy**
>
> Thanks for raising this.
>
> The distortion metric in the FAM, and this paper are actually different.  We clarified this  in the revised paper.
>
>
> In the FAM paper, the distortion is "the average of the **absolute angle differences (AAD)** between each corresponding angle of the 3D triangles and the 2D parameterized triangles." (See the 3rd line from the bottom on page 7 of the FAM paper (https://arxiv.org/pdf/2405.14633) )
>
>
> | **AAD metric** | | | | | | | | | | | | | | |
> | ------- | :-----: | :-----: | :-----: | :--------: | :-----: | :--------: | :-----: | :-----: | :-----: | :------: | :-----: | :-----: | :-----: | :-----: |
> | Model | Bimba | Lucy | Ogre | Armadillo | Bunny | Nefertiti | Dragon | Homer | Happy | Fandisk | Spot | Arm | Cow | Average |
> | FAM | 0.064 | 0.166 | 0.227 | 0.154 | 0.073 | 0.038 | 0.239 | 0.048 | 0.175 | 0.087 | 0.080 | 0.040 | 0.106 | 0.115 |
> | SeamGPT | 0.000 | 0.011 | 0.009 | 0.008 | 0.000 | 0.001 | 0.184 | 0.001 | 0.188 | 0.000 | 0.003 | 0.000 | 0.003 | 0.031 |
>
>
>
>
>
> In this paper, we compute the **Per-triangle Conformal Energy (PTCE)**. This is done by computing the deviations of singular values in the 3D-2D Jacobian matrix (c.f. Appendix section C). This is an established metric for measuring triangular distortion.
>
>
>
> | **PTCE metric** | | | | | | | | | | | | | | |
> |------|------|------|------|---------|-------|---------|--------|-------|-------|--------|------|-----|-----|------|
> | Model | Bimba | Lucy | Ogre | Armadillo | Bunny | Nefertiti | Dragon | Homer | Happy | Fandisk | Spot | Arm | Cow | Average |
> | FAM  | 12.10 | 35.14 | 11.55 | 59.87 | 7.33 | 11.21 | 904.89 | 14.19 | 23.00 | 12.21 | 9.37 | 20.98 | 8.49 | 86.95 |
> | SeamGPT | 10.68 | 0.01** | 2.01 | 2.47 | 50.47 | 0.12 | 0.56 | 10.28 | 61.68 | 8.15 | 5.95 | 14.88 | 2.24 | 13.04 |
>
>
> Under both metric, SeamGPT shows lower distortion than FAM.
>
> The visual results in Figure 3 also prove this: UVs from FAM are visually more distorted.

---

> > ### Comment · Area_Chair_bwbJ · 2025-11-24
> >
> > Thank you for the clarification. However, I find the explanation unconvincing. On identical 3D models, the visual results you report for FAM are noticeably worse than those presented in the original paper.

---

> > > ### Author Response · Authors · 2025-11-24
> > > **Visualization bug fix**
> > >
> > > We just fixed the visualization BUG now the visual results closely resembles the results in the original  FAM paper. We updated Figure 3 accordingly.
> > >
> > > The default visualization code in FAM output very sparse UV maps, which makes it looks very bad. Adding "--N_poisson_approx" arguments fixed this problem.  (check the testing command in https://github.com/keeganhk/FlattenAnything/blob/master/README.md)
> > >
> > >
> > > Since this was done using the same pre-trained model, it does not affect the distortion metric results.
> > >
> > > Overall, the visualization of FAM's UV map indicates that the results are heavily distorted.
> > >
> > > The two models in Figure 3, Nerfititi and Fandisk, also appear in the original FAM paper. We did observe a slight difference in the visualization for Fandisk: our results show more seam cuts than in the original FAM paper. The result for Nerfititi is nearly identical to original FAM paper.
> > >
> > >
> > > Please note that we **ran the open-source code of FAM without any modifications**.

---

### Author Response · Authors · 2025-11-27
**Looking forward to the discussion**

Dear Reviewers,

Thank you again for the time and effort you have devoted to reviewing our submission. We have carefully addressed the questions and concerns raised across the reviews, and the updated manuscript and supplementary material now incorporate all corresponding revisions.

We sincerely appreciate your thoughtful feedback and would be grateful for any further comments or assessments you may have as we move forward in the discussion phase.

Best regards,
The authors

---

### Meta-Review · Area_Chair_wGmq · 2026-01-06

**Summary:**

The primary concerns raised by reviewers revolve around three key issues: the fairness and validity of comparisons with the FAM baseline, the limited technical novelty of applying auto-regressive modeling to surface cutting, and insufficient empirical evidence to support claims of exceptional performance. Specifically, the Area Chair and multiple reviewers questioned whether the reported FAM results were fairly represented, noting that visual outputs appeared worse than in the original paper. Reviewers also argued that the method largely adapts existing components (e.g., point cloud encoders, GPT-style decoders) without substantial innovation. Additionally, while the method reduces fragmentation, its distortion metrics are often comparable to or worse than non-learning baselines like XAtlas, undermining claims of overall superiority.

**Reviewer Concerns:**

The rebuttal partially addressed the metric discrepancy with FAM by clarifying differences in distortion calculations and fixing a visualization bug. However, the explanation did not fully resolve concerns about the fairness of the visual comparison, as the Area Chair remained unconvinced. The authors defended the novelty by framing the problem reformulation as a conceptual contribution, but this did not alleviate reviewer skepticism about the incremental nature of the technical approach. Concerns about the method’s sensitivity to quantization, seam-length control, and generalization to complex topologies were acknowledged but not conclusively answered. Overall, the core issues of comparative fairness and limited technical advancement remain outstanding.

**Reviewer Scores:**

**Reviewer nRhE (initial score: 10)**: Likely would lower the score significantly, given their expressed willingness to re-evaluate based on the FAM comparison issue, which was not fully resolved.

**Reviewer fdf2 (initial score: 6)**: Would likely maintain a borderline score (~6), as they previously noted the paper’s limited impact and remained unconvinced by the rebuttal regarding novelty.

**Reviewer 5xei (initial score: 4)**: Might slightly lower or maintain their score (~4), as they found the rebuttal defensive and noted unmet suggestions, such as including relevant citations or adequately addressing UV representation drawbacks.

**Reviewer 4sRE (initial score: 6)**: Would probably retain a score of 6, as their concerns about incremental contribution and mixed empirical results were not substantially alleviated.

---

### Decision · Program_Chairs · 2026-01-26

Reject